# Wine Industry Waste as a Source of Bioactive Compounds for Drug Use

**DOI:** 10.3390/ijms262210820

**Published:** 2025-11-07

**Authors:** Mariana Mesta-Corral, Nathiely Ramirez-Guzman, David Aguillón-Gutiérrez, Cristian Torres-León, Jorge Aguirre-Joya

**Affiliations:** 1School of Biological Science, Universidad Autónoma de Coahuila, Torreón 25280, Mexico; mestam@uadec.edu.mx (M.M.-C.); nathiely.ramirez@uadec.edu.mx (N.R.-G.); 2Research Center and Ethnobiological Garden, Universidad Autónoma de Coahuila, Viesca 25280, Mexico; david_aguillon@uadec.edu.mx (D.A.-G.); ctorresleon@uadec.edu.mx (C.T.-L.)

**Keywords:** winery wastes, bioactive compounds, fermentation, natural drugs

## Abstract

Wine is one of the oldest alcoholic beverages, produced from the fermentation of the grape *Vitis vinifera*. Currently, the wine industry is exploited worldwide in multiple regions, generating significant amounts of agro-industrial waste at different stages of the production chain. These wastes represent a source of environmental contamination due to the toxic nature of some of their compounds. From a biotechnological perspective, the by-products of the wine industry are an attractive source of bioactive compounds with potential applications in various fields, including food, pharmaceuticals, and cosmetics. The extraction of these compounds can be carried out using fermentation techniques that utilize microorganisms to facilitate the release and biotransformation of the desired metabolites through their enzymatic tools. This work provides a review of the history of the wine industry and its current activities, describes the wine production process, and outlines the waste generated during this process. The fermentation process is described as a biotechnological alternative for the valorization of these residues. This purpose enables their reintegration into the production chain through the extraction of high-value bioactive compounds with potential use as drugs in pharmacology.

## 1. Introduction

Wine is one of the oldest alcoholic beverages in the world. This drink is made from the fermentation of *Vitis vinifera* grapes, of which multiple varieties are spread around the world, each with unique aromatic characteristics and flavors. Currently, the wine industry is developing in multiple countries worldwide. According to data from the International Organization of Vine and Wine (OIV), it is estimated that the total area worldwide dedicated to vineyards is approximately 7.2 million hectares [1]. In 2022, grape production was recorded to have reached 74.9 million tons, with approximately 71% being used for wine production [2]. In 2023, wine production worldwide reached 237 million hectoliters, driven mainly by France, Italy, Spain, the USA, and Chile [1].

Throughout the wine production chain, significant amounts of waste are generated in a short period, equivalent to approximately 30% (*w*/*w*) of the initial grape weight [3]. These wastes include vine shoots derived from pruning, stems, grape pomace, seeds, lees, and wastewater. The improper management or disposal of these wine wastes poses significant economic, social, and environmental problems due to the large volumes generated. Other factors that contribute to the management challenge include their content of heavy metals, pH, salinity, and the polyphenolic compounds that render them naturally phytotoxic [3,4,5].

Current international political trends, driven by the United Nations’ Sustainable Development Goals and the 2030 Agenda, point towards the development and strengthening of a circular economy, characterized by the revaluation of waste [6].

The waste generated by the wine industry represents a valuable source of bioactive compounds with applications in various areas, including food, cosmetics, pharmaceuticals, and energy, among others [7].

The application of bioprocess techniques to extract bioactive compounds represents an ecologically and economically sound methodology for bioactive recovery [8].

The use of microorganisms for extracting bioactive compounds from agro-industrial waste through fermentation processes is a widely studied biotechnological technique. In this process, the enzymatic machinery of microorganisms that develop on the substrates is utilized, using them as a source of nutrients and degrading the waste to obtain these compounds, while allowing for the extraction and biotransformation of the compounds of interest [9,10].

Our paper provides an overview of the history behind the wine industry and the current context, describes the wine production process, highlights the waste generated during the production stages, and presents fermentation as a biotechnological alternative for the revalorization of wine industry waste, allowing its reinsertion into the production chain through the extraction of bioactive compounds with high added value.

## 2. Winery Industries

### 2.1. Historical Context

Wine is one of the oldest alcoholic beverages in the world, with its ancient traces located in the Caucasus region near 5000–6000 BCE., parallel in Georgia and the North Zagro Mountains, both placed in a natural habitat of *Vitis vinifera*, the evidence is related to potteries where tartaric acids residues are identified, being this organic acid a particular component in grapes and wine, but rarely in other fruits [11,12]. This product spread to the south in all Middle Eastern regions. Despite this, each culture has its own history of wine production, which is related to spontaneous fermentation that occurs after grapes are stored, similar to other fermented foods, and has evolved [13]. Ancient civilizations had records of vineyards; nevertheless, the Egyptians developed structured techniques in wine making, including harvesting, pressing, crushing, and storage, that are well-documented [14]. The rest of the syringic acid and malvidin 3-o-glucoside were detected in Egyptian recipients of white and red wine, both of which are molecules derived from wine [12]. Wine consumption is regarded as having medicinal, religious, and nourishing properties, and is especially associated with wealthy societies.

Greeks have developed an extended methodology to improve wines, such as retarding harvesting to produce a sweeter drink. The consumption of wine by Greeks was linked to both gastronomic and social purposes, and occasionally to medical activities. Sometimes, wines were mixed with a variety of spices and aged for years; they produced a few dry white wines. Greeks promoted the winery through each of their conquered colonies along the Mediterranean Sea [15,16,17].

The Roman Empire was the next significant promoter of wine making in Europe. Through their extended occupation, Romans expanded vineyards across Europe to supply the Roman garrison, also importing barrels [17]. Huge amounts of information on viticulture (grape variety classification, disease identification preferences, irrigation, and fertilization) were imparted by the Rans [16]. In many European regions, wineries already existed (Spain, Portugal, the South of France, Bulgaria, and Turkey, Romania), and the Romans only perfected their techniques. However, in parts of England and North-Central France, the first vineyard reports are found after the Romans [17].

Then, Cistercian monks began to cultivate and improve European wine, especially in French Regions, developing for the first time the term “terroir” and restoring Europeans’ staple drink. However, growth was not linear due to some climatic changes [17].

After the expansion of European Kingdoms, *Vitis vinifera* arrived in America, and the wine-making process, with Franciscans and Jesuit monks, led to the first wine production in Mexico, which then spread to South America. By around 1500 CE, many of these viticulture settlements remained productive until the present [17].

In the United States, the reproduction and establishment of *V. vinifera* was difficult, especially across the East Coast, until plant crossing with native strains occurred. However, by 1700, the winery level in California, specifically on the West Coast, was comparable to French wines. Simultaneously in Europe, other attachments were incorporated into wineries, including glass bottles and cork. At the same time, the Dutch introduced viticulture to South Africa and Australia. In 1900, Pasteur elucidated the fermentative processes in wine production, and in the years that followed, the worldwide winery industry dedicated itself to accelerating automation and improving processes to what they are today [14,16,17].

### 2.2. Winery Procedure

Over time, wine manufacturing has evolved, and with this, numerous technologies have been incorporated. Nevertheless, a group of key steps has been maintained to the present day.

#### 2.2.1. Destemming

The first step in winery production is grape harvesting, after which vegetal material in the grapes, including leaves, stems, and stalks, is removed [18]. The removal could be avoided after crushing works in a low-polyphenolic content vine variety, resulting in enhanced color and body. Nevertheless, stems can also add or elevate the content of diverse fatty acids and methoxypyrazines, which mature into unpleasant odors during the fermentative stage [19,20]. Leaves and other green parts could therefore lead to an elevated concentration of quercetin, which generates precipitation or undesirable haze in red and white wines [11,21].

#### 2.2.2. Crushing

The grapes are crushed to extract the sugary juice from the fruit and prevent contamination and oxidation reactions [18]. The crushing procedure involves passing grapes through a perforated cylinder or well, which is pressed by rolls. The employed rolls could be adjusted to produce different press levels, allowing for adaptation to multiple grape varieties [11]. Other alternative methods for grape crushing exist, including the application of centrifugal force; however, a fruit slurry is generated, which complicates the clarification process. Novel crushing techniques were explored, including Cryoextraction and supraextraction. During this technique, grapes are frozen and then maintained at lower temperatures, causing cell rupture, which facilitates more successful pressing [22].

During grape crushing, it is crucial not to break the seeds, as they are primarily a source of oils and catechins, and their incorporation into the juice also generates bitter, herbaceous, and rancid odors [11].

#### 2.2.3. Maceration

During the maceration procedure, the contact between solids and liquid components is extended. The chemical composition enables the release of chemical compounds bound to the grape skin, flesh, or seeds, including polyphenolic compounds. The maceration procedure is extended by different time lapses, depending on the wine variety [18,22]. Maceration can be achieved under various conditions, including temperature control (4–28 °C), applied before, during, or after fermentation, the presence of oxygen, and the duration of contact (2–48 days), which differs for each class of wine. Normally, red wines require extended maceration time, while rosé or white wines demand low or no grape skin exposure [23,24].

#### 2.2.4. Pressing

Pressing allows for the separation of grape liquids from solids. Especially in elevated volumes, a de-juicing is carried out prior to pressing. The de-juicer consists of columnar tanks with perforated baskets at the end, allowing for juice extraction and retaining grape pomace. The resulting pomace is normally pressed to extract all the liquids [22]. Press fractions are rich in tannins and gums, as well as in organic acids, aroma compounds, ions, and proteins [25,26].

#### 2.2.5. Must Preparation

Before fermentation, the must needs to be prepared by adjusting the pH, performing thermovinification, adding sulfur dioxide, diammonium phosphate, and thiami as well as chaptalization and blending (free-run-press-n mixtures) [22,23]. All these activities are designed to ensure an adequate environment for winery yeast work.

#### 2.2.6. Alcoholic Fermentation

The primary step in winemaking involves the transformation of sugars into alcohol, primarily ethyl alcohol, by the yeast *Saccharomyces cerevisiae*. The fermentative process is developed in stainless tanks or oak barrels for Chardonnay varieties [18]. *S. cerevisiae* metabolism is adapted to alcoholic fermentation. This microorganism exhibits an elevated rate of sugar bioconversion, a feature regarded as highly efficient due to its alcohol dehydrogenase activity, elevated levels of glycolytic enzymes in the cytoplasm, and a lack of respiratory enzymes in the mitochondria until non-fermentable compounds appear or prevail in the medium culture [22]. The alcoholic fermentation could be extended for several days, depending on the must composition. This process involves the release of energy in the form of heat, necessitated by the need for temperature control, which ranges between 20 and 25 °C. Yeast also produces other classes of metabolites, including higher alcohols, organic acids, and polyols, which contribute to the character of theine [27].

#### 2.2.7. Malolactic Fermentation

Grapes and consequently must be rich in organic acids, including citric, tartaric, and malic. Additionally, during alcoholic fermentation, a significant amount of organic acids is produced by yeast and bacteria. All these molecules contribute to the organoleptic properties of wine and impact its stability [28]. To improve this issue, a second fermentation is executed. The secondary fermentation in wine is malolactic and driven by Lactic Acid Bacteria. Various genera are involved, including Lactobacillus, Pediococcus, and Leuconostoc, but *Oenococcus oeni*, dominates the process [29,30].

During this secondary fermentation, the harsh and highly concentrated malic acid is bio-transformed by LAB into a softer lactic acid, which contributes to the wine’s complex flavor, increases its pH, and promotes microbiological and physicochemical stability [23,30].

#### 2.2.8. Clarification

White wines are typically clarified before the fermentation process. The must suspended particles, lead to the formation of “fusel alcohols” (non-ethanolic) and advocate the activity of polyphenol oxidase and hydrogen sulfide production [22]. Clarification can be achieved through various methods, such as decanting and racking, filtration, or centrifugation [18].

A group of components could be employed for clarification procedures, including proteins, clays, synthetic polymers, and activated carbon, among others [18].

#### 2.2.9. Maturation

The oak species employed by their manufacturers include Quercus petraea and *Quercus robur* (French or European oaks) or *Quercus alba* (American oak) [31]. Normally, these barrels of oak possess a toasted surface and act as an active vessel for 9 to 22 months, which is very important in red wines [23]. During the maturation process in the barrel, multiple reactions are involved. The fermented liquid is transformed into a very complex product. Firstly, the barrel releases volatile compounds into the wine, contributing to the wine’s aroma. Additionally, they discharge ellagitannins, which help ameliorate the mouthfeel. Low oxygen concentrations in wine promote polymerization reactions among anthocyanins and proanthocyanidins, which generate color stabilization and reduce astringency. Finally, during the maturation stage wine generates precipitates, resulting in a limpid wine [23,31]. To withdraw the sediments, activities like racking could be employed at this stage.

#### 2.2.10. Filtration

Before bottling, wine demands refining activities, including clarification procedures previously mentioned (Clays, proteins, etc.), as well as racking or centrifugation are also possible. After that, a filtration step is needed. Filtration is possible by employing various techniques, including earth filtration, sheet filtration, rotary vacuum filtration, or membrane filtration. Filtration avoids hazel formation in wines [23].

#### 2.2.11. Stabilization

During this step, wine is cooled in vats at −4 to 0 °C, allowing tartrate crystals to precipitation [23].

#### 2.2.12. Bottling

Previously, wine was filtered or sterilized by different pasteurization techniques. Bottles are closed employing cork, a natural vegetal product derived from *Quercus suber* bark [23]. Cork allows gas transfer between wine and the exterior and thus enables a range of chemical reactions (polymerization, hydrolysis, oxidation, reduction) to occur in wine. Wine continues to mature in bottles, contributing to its flavor [32,33]. The main steps of the winery industry are shown in Figure 1.

### 2.3. Actual Wine Context

The International Organization of Vine and Wine (OIV) reports a worldwide vineyard area for 2023, estimated at 7.2 million hectares, including areas for wine, table grapes, and grapes to be dried. Being the countries with the most extensive vineyard cultivars, Spain, France, China, Italy, and Türkiye, together represent 50% of the overall vineyards [1]. For America, the main producers are the USA, Argentina, Chile, and Brazil, representing 11.7% [1]. The overall grape production in 2022 reached 74.9 million tons [2], and it has been reported that nearly 71% of grapes are used for wine production [34].

Wine production in 2023 was 237 million hectoliters, with the main producers being France, Italy, Spain, the USA, and Chile, which account for 63% of the overall production [1]. While major consumers include France, the USA, and Italy, the leading consumers per capita are France, Portugal, and Italy. The average consumption per capita at the global level is around 3.53 L per year, and many countries exceed this value by up to 14 times [35].

Mexico is one of the oldest wine producers in Latin America, and over the last eight years, consumption per capita has increased to 1.2 L. As of 2021, there are nearly 250 warehouses in operation, producing around 36 million liters per year from 73,000 tons of grapes [36]. At least 75% Mexican winery industry is concentrated in Baja California, and the rest is distributed among Zacatecas, Coahuila, Queretaro, Aguascalientes, Sonora, and Chihuahua [37], with cultivars in at least fourteen Mexican states and generating above 500,000 jobs, with economic remunerations of $ 24.8 million per year [36].

At the global level, in 2023, a revenue of $330.1 billion was estimated from wine production [38]. Despite this, the big picture in the wine world shows that last year’s wine production decreased by 11.6% compared with 2022. This could be linked especially to global climatic adversities and vineyard susceptibility. Additionally, wine consumption is low, at around 2.6%, particularly attributed to geopolitical conflicts, energy, and global supply crises, resulting in rising prices [1].

The wine industry represents a historically well-established sector; nevertheless, it faces multiple changes, particularly in terms of climatic and economic challenges. The winery’s current state enables revisions to its contributions to a circular economy and environmental care issues related to the process.

## 3. Winery Industries By-Products

The amount of waste generated by the wine industry depends on the processes used at each winery. It is generated both during the pre-production stages (agricultural) and during the wine production process. During the first stage, waste called vine shoots (made up of leaves and tendrils derived from pruning) is generated, while during the production process, waste is generated consisting of lees and grape pomace (pulp, seeds, stems, and fruit skin) [4].

### 3.1. Vine Shoots

Vine shoots represent the most abundant residue generated in vineyards [39]. It has been reported that the production of vine shoot residues ranges between 2–5 Tons/ha annually, depending on the climate, plantation density, and vine vigor, among other agronomic factors [4]. Its dry base composition has been reported to comprise 39.7% cellulose, 22.7% hemicellulose, 21% Klason lignin, 3.4% protein, 2.7% ash, and 0.3% lipids [40].

These residues have low economic value and have been used by farmers for years to enrich the soil as compost due to their rich protein and mineral content. However, in some cases, they are incinerated to avoid contamination by fungi [41,42].

Novel applications of these materials include the production of activated carbon, solid biofuels, and paper pulp, as well as the preparation of substrates for edible mushrooms, foliar additives, and oenological additives in the form of wood chips [43,44].

### 3.2. Grape Pomace

Pomace is the most abundant residue in the wine industry, accounting for approximately 60% of the waste generated. Pomace is obtained from the pressing of grapes and is composed of the skin, seeds, and some remnants of stems. This residue accounts for approximately 15% of the grape, and its composition depends on the variety, type of processing, and cultivation conditions [4,44].

For every 6 L of wine, approximately 1 kg of pomace is generated [45]. A ton of grape pomace is composed of approximately 425 kg of skin, 225 kg of seeds, 249 kg of stems, and other minor compounds such as water [46]. In the production of red wine, the entire grape undergoes fermentation, whereas in the production of rosé and white wine, only the juice is fermented, resulting in variations in the composition of the generated pomace. This is a residue rich in fiber, phenolic compounds, minerals, and colorants [47]. Grape pomace is composed proximally of protein (3–14%), fat (1–14%), fiber (17–88%), carbohydrates (12–40%), and ash (1–9%) [47]. It also has a moisture content of 50–72% depending on the variety and degree of ripeness, and a lignin content of 16–24%. Pectin substances are the main component of the cell wall of grapes, with a content of 37–54% of the total cell wall polysaccharides, with cellulose being the second polysaccharide that makes up the cell wall with a content of 27–37% [46]. Traditionally, this residue is used by farmers to enrich the soil as fertilizer or for animal feed [45].

### 3.3. Wine Lees

Wine lees are residues that accumulate at the bottom of the tanks after alcoholic fermentation. They are composed of yeast cells, bacteria involved in the wine making process, and other insoluble particles. Wine lees are rich in polysaccharides, lipids, and proteins, among other organic compounds of interest such as ethanol, tartaric acid, and polyphenolic compounds, and represent about 25% of the residues generated during wine production [41,48].

Wine lees can be classified in different ways. An example of this is the first and second fermentation lees, which are the lees generated during the alcoholic and malolactic fermentation processes, respectively. There are also the aging lees, produced, as their name suggests, during the aging process of wine in wooden barrels. On the other hand, they can also be classified according to their particle size, divided into heavy lees (2 mm–100 µm) and light lees (<100 µm) [49]. Commonly, these residues are discarded into the environment; however, due to their composition rich in organic matter of 900–35,000 mg/L, chemical oxygen demand of 30,000 mg/L, and high content of polyphenolic compounds whose nature is phytotoxic, their careless disposal without adequate treatment can result in serious damage to the transpiration cycles and photosynthesis processes of plants, also reducing soil fertility [5]. A graphical resume of winery industry by-products is presented in Figure 2.

### 3.4. Winery By-Products and Circular Economy

According to United Nations, the need for resources is expected to arise in the coming decades. Thus, the “linear economy” model, which contemplates the disposal of waste at the end of the processes, becomes obsolete, and the “circular economy” model becomes an ideal. The circular economy is defined as a model in which the value of resources, materials, and products is maintained as much as possible and the generation of waste is minimized [44,50]. In the circular economy model, sustainability-oriented ideals take place through the revalorization of waste by incorporating it back into the commercial value chain through the incorporation of innovative techniques, rigorous planning, and strategic management [44], making use of the three “R”s (reduce, reuse, and recycle for the development of sustainable processes [50].

The waste generated by the wine industry represents between 31% and 40% of the total grapes harvested [5].

Although in many regions the waste generated by wine manufacture, is traditionally used as fertilizer or to obtain liquor, from the point of view of the circular economy, they represent an attractive source of bioactive compounds (rich in fiber, protein, lipids, organic acids, phenolic compounds, and lignocellulosic materials) with diverse applications, offering the possibility of revalorizing them through the generation of compounds with added value [51].

## 4. Natural Drugs from Wine Industry Waste Through Fermentation

Fermentation processes are biotechnological processes that consist of the transformation of a substrate through the action of microorganisms (mould, bacteria, and yeasts). These are capable of extracting compounds present in the substrate through the action of enzymes, which, in turn, can carry out the biotransformation of these compounds [52]. Fermentation processes are divided into two systems, solid fermentation and liquid fermentation. Solid fermentation is characterized by being developed on substrates or supports in the absence of free water. It stands out for its versatility and low technological requirements for its development, in addition to the fact that it does not require high-level technical training for its execution. Due to its characteristics, it is a system commonly used to work with filamentous fungi [53]. Liquid fermentation, on the other hand, is a system in which the substrate is dissolved in a liquid medium in a homogeneous manner. Although this type of fermentation has greater technological demands than its counterpart and poses a higher risk of process contamination due to its high liquid content, it offers the advantage of controlling variables throughout the process development. Due to its characteristics, this system is commonly used when working with yeasts and bacteria [54,55].

Multiple investigations report the use of fermentation processes for the revalorization of different agro-industrial residues such as mango [56], pineapple [55], coffee [57,58], green tea [58], grape pomace [59], tomato [60], pomegranate [61] and rambutan [62], to name a few, by obtaining different products such as polyphenolic compounds [52], enzymes [58,63,64], biopesticides [53], biofuels [65]. Similarly, there are reports of using fermentation processes to obtain bioactive compounds of interest through the valorization of waste from the wine industry, a compilation of which is presented below.

### 4.1. Polyphenols

Polyphenolic compounds (PC) are a group of secondary metabolites synthesized by plants. In nature, PC serve as pollinators attractants, microbial, herbivorous, and UV radiation protection [66]. The structural units in PC possess at least one phenolic ring and one or more hydroxyl groups. These molecules can range from a single monomer to an elevated number of monomers, thereby increasing their complexity [67]. Depending on the key structures and grade of complexity, PC could be mainly classified into phenolic acids, stilbenes, flavonoids, and tannins [68]. They are found in many plant-derived foods and contribute to various organoleptic features, such as color, odor, and astringency [67]. They exhibit important bioactivities, and it has been reported that their consumption could prevent various human conditions related to cardiovascular, cancer, aging, and diabetic or inflammatory diseases [66,67].

Grapes are not an exception, and therefore, wine is a beverage rich in these molecules, which not only contributes to the organoleptic experience of drinking wine, but also to health, remembering “The French paradox” [16]. The PC concentration in wine varies among employed grape varieties; however, in general terms, red wines (1–5 g/L) are richer in these molecules than white wines (0.2–0.5 g/L) [24]. These important compounds are normally found in grape peels, seeds, or stems and are released into the beverage throughout all the process steps. Nevertheless, a considerable amount is retained in by-products, considering that only 30–40% of PC is extracted into wine [69].

#### 4.1.1. Flavonoids

Flavonoids are a group of compounds with a C_6_-C_3_-C_6_ nucleus; they possess a highly diverse classification depending on their hydroxylation pattern. They could be divided into Flavonol, Isoflavones, Flavones, Flavonols, Flavonones, Flavononols, and Anthocyanins [67,68].

Anthocyanins

Anthocyanins are a type of flavonoid, structurally characterized by having a 15-carbon skeleton (C_6_-C_3_-C_6_) formed by two aromatic rings (A and B) connected through a three-carbon bridge, which gives rise to the formation of a heterocyclic ring (C). This structure allows a diversity of substitutions and modifications, which determine their antioxidant and biological properties [59]. Anthocyanins are natural, water-soluble pigments which are present in the vacuoles of plant cells in flowers, fruits, leaves, roots and stems [70]. These compounds are secondary metabolites that belong to the polyphenol group and are the primary determinants of red and blue colors and their derivatives in plants. In addition, they play an important role in pollination, seed dispersal, and organ development in plants, as well as in their adaptation to abiotic changes such as droughts, high light intensity, and nutrient and biotic scarcity, including pathogen attacks [71]. Anthocyanins have a wide variety of forms and can change their structure and color in response to pH changes. These compounds can be extracted from various plants, including red roses, blueberries, red cabbage, purple sweet potatoes, black bean seed coats, black carrots, eggplants, roselle, mulberries, and orchid ginger [72]. The anthocyanin content in plants varies among different species and is primarily influenced by genotype, with a lesser extent by climatic conditions, light exposure, maturity, and storage conditions [73].

As mentioned above, anthocyanins are found within the vacuoles of plant cells; therefore, their extraction requires the disruption of these cellular structures. This can be achieved through fermentation processes, thanks to the enzymatic action of microorganisms. Amaya-Chantaca et al. reported [56] the extraction of polyphenols through solid and liquid fermentation processes using grape pomace as a substrate, employing a strain of *Aspergillus niger* GH1. In their research, they determined that solid fermentation allowed a greater extraction of compounds, reaching titers of 2.262 g·L^−1^ of hydrolysable tannins and 3.684 g·L^−1^ of condensed tannins, compared to liquid fermentation supplemented with salts, where they achieved titers of 0.182 g·L^−1^ of hydrolysable tannins and 0.198 g·L^−1^ of condensed tannins. Through the development of enzymatic kinetics, they demonstrated that the extraction of polyphenolic compounds, among which are anthocyanins, is associated with the action of the enzymes tannase, ellagitannase, β-glucosidase, xylanase, and α-L-arabinofuranosidase. Among the extracted anthocyanins, they identified Cyanidin 3-O-xylosyl-rutinoside, Pelargonidin 3-O-rutinoside, Cyanidin 3-O-(6″-acetyl-glucoside), Cyanidin 3,5-O-diglucoside through the application of HPLC-MS technologies.

Among their bioactivities, anthocyanins contribute to alleviating metabolic dysfunction induced by obesity and diabetes, offering protection against their recurrence. They could interact with intestinal glucose transporters, interfering with their absorption. Their intake helps modulate regulatory pathways, such as insulin signaling, adiponectin secretion, and increased AMPK, which increases GLUT 4 protein levels and its translocation to the plasma membrane in various tissues. In addition, through the stimulation of SIRT1, AMPK, and PGC1 and thanks to their antioxidant activity, they promote mitochondrial density and function in adipose and skeletal muscle tissues [74]. These compounds have been reported to exhibit anticancer activity, which is attributed to the catechol structure in the B-ring. The ability to eliminate free radicals helps prevent the spread of cancer cells and thus the possibility of metastasis. It has been reported that an adequate dose of anthocyanins administered through food can inhibit tumor inflammatory reactions by regulating the expression of cyclooxygenase-2 (COX-2), inducible nitric oxide synthase, and nuclear factor kappa-B (NF-κB), thereby inhibiting the proliferation of tumor cells [75].

There is evidence that anthocyanins confer a protective effect against neurodegenerative diseases. They help maintain memory and contribute to the relief of cognitive function alterations by protecting neurons, hippocampal nerve cells, and glial cells against damage from glutamate, Aβ, and lipopolysaccharides. Additionally, they reduce inflammation, oxidative stress, and apoptosis in nerve cells [76]. The consumption of these compounds is also associated with changes in the intestinal microbiota, which favors the development of anaerobic bacteria associated with reduced oxygen levels in the gastrointestinal lumen and a reduction in oxidative stress [77]. On the other hand, it has also been reported that their consumption influences anti-osteoporotic bone resorption by suppressing osteoclast formation [78].

Catechins

They are a group of Flavan-3-ol-type of flavonoids. In grape peel and seeds, Epigallocatechin 3-gallate, Epigallocatechin, Gallocatechin, (+)-Catechin, and (−)-Epicatechin are commonly found [24], and some reports also indicate their presence in grape stalks [41]. They are the most abundant flavonoids in grapes and wines [79].

Some authors employ a FES using a co-culture of Aspergillus niger GH1 and Pichia strains over grape pomace and observe that from 72 h of culture, (+)-Catechin is released [59]. Also, ref. [80] used Othello black grape pomace for an FES using the fungi Rhizopus mini, they obtained concentrations of (+) Catechin, and (−)-Epicatechin 3.49 ± 0.52 and 1.37 ± 0.08 mg/100 g, respectively, at the seventh day of fermentation, in a pretreatment the author show that cellulases derived from the same fungus also enhance catechin concentration.

Tannins

Tannins are a group of PCs that confer astringent mouthfeel in wine. They are divided into hydrolyzable tannins and condensed tannins. Grape naturally possesses the condensed type [81]. Hydrolyzable tannins are a group of elevated molecular weight formed by monomers. There are two types of gallotannins (gallic acid) and ellagitannins (ellagic acid). They are susceptible to hydrolysis by pH changes or enzymatic procedures. Normally, they are not present in grapes, except in muscadine grapes, but are really important in wine through barrel aging [24].

Condensed tannins, also known as procyanidins, are the PC most abundant in grape seeds [82], representing as much as 75–85% of condensed tannins [83], but are also reported in peels and pulp. They are polymeric compounds that produce anthocyanidins, and their polymerization grade varies depending on their tissue location [81]. These molecules increase during wine aging and tend to form precipitates. They are found naturally at concentrations between 1.2–3.3 g/L [24]; nevertheless, nearly 50–75% of them remain in grape pomace [82].

Grape pomace is employed [52] derived from wine production and compares FES and SumF employing A. niger GH1, showing 3.684 g·L^−1^ of condensed tannins for FES and 0.198 g·L^−1^ for SumF. Authors link these activities to a group of enzymes, including β-glucosidase, xylanase, α-L-arabinofuranosidase, and tannase.

#### 4.1.2. Resveratrol

Resveratrol (RSV) is a PC from the stilbene class (3,5,4′-trihydroxy-stilbene), normally found in *V. vinifera* leaves epidermis, grape skins, and seeds [83], known as phytoalexin and responsible for antimicrobial response against the pathogen in plants [84]. RSV is one of the most important PC derived from grapes. Through the winery, only 11% of RSV was extracted, making pomace, a by-product, an important source of this molecule [82].

RSV supply is narrow due to elevated production costs [85], and bioprocessing for RSV release in winery by-products appears to be an emerging alternative; nevertheless, it remains underexplored.

Ref. [86] employed three different fungi (*Aspergillus niger*, *Monascus anka,* and *Eurotitum cristatum*) for FES on grape-pomace seeds, under 28 °C and 65% humidity. *M. anka* shows an increase of 6.42 times in total phenolic content, including RSV. Additionally, ref. [85] employ an immobilized microbial consortium comprising yeast, *Aspergillus oryzae*, and Aspergillus niger for RSV extraction from grape seed residues pretreated with a surfactant-ionic liquid, achieving 305.98 ± 0.23 μg/g of RSV. Authors [87] analyze the potential of three winery by-products for resveratrol production through a techno-economic and environmental study. The employed residues were grape pomace from pressing, lees, and vine pruning residues. They employed a fermentative procedure using an engineered *S. cerevisiae* that produces RSV from some sugars. If this procedure does not imply an extractive methodology, it can utilize these by-products as an effective culture medium in fermentative procedures, where pretreatments are part of the technical challenges. Additionally, ref. [88] demonstrates how yeast *Hanseniaspora uvarum* exhibits great potential for producing wines with elevated RSV levels, being regarded as a β-glucosidase producer that facilitates the hydrolysis of resveratrol-glucosides. Being a potential alternative for the RSV in residues.

RSV possesses multiple health-related functionalities, including anti-carcinogenic, cardiovascular protective, and antioxidant properties [82].

#### 4.1.3. Phenolic Acids

Phenolic acids represent a non-flavonoid PC. Phenolic acids could be separated as Hydroxybenzoic acids (HBAs) with the benzoic ring and a carboxylic acid group (C_6_-C_1_), and hydroxycinnamic acids (HCAs) with the same core structure but differences in the carboxylic acid ramification (C_6_-C_3_) [24]. In grapes and wines, the most abundant HBAs are vanillic, gallic protocateuchuic, hydroxybenzoic, and syringic acids, while the main HCAs include ferulic, sinapic, p-Coumaric, and caffeic acids [81]. Phenolic acids are found in grape vacuoles located in peels and pulp cells. Normally established as sugary, esterified, or organic acid-type forms. Approximately 20–25% remains in a free form, facilitating leakage to wine [82].

### 4.2. Alcohols

Wine industry waste can be used to obtain alcohol through fermentation processes. Kartpe et al. reported the production of xylitol and propanol using wine industry waste through solid fermentation processes using a strain of *Penicillium chrysogenum*. In the case of xylitol, titers of up to 24 mg/L were obtained during the last phase of fermentation (eighth day), while in the case of propanol, titers of up to 6.2 mg/L were obtained during the lag phase (sixth day) [89]. For their part, Salgado et al. reported the revalorization of vinases through the production of an economic means for obtaining xylitol through fermentation processes with *Debaryomyces hansenii* reaching titers of up to 33.4 g/L [90]. Xylitol is a polyol used as a sweetener, which is believed to have antiplatelet, gingival inflammation-reducing, and anti-cavity effects because it reduces the growth levels of *Streptococcus mutans* and *Streptococcus sangui* and also reduces the production of bacterial β-glucosidase activity in human saliva, which is essential for the formation of biofilms in the oral cavity. Furthermore, thanks to its ability to bind calcium, it contributes to the remineralization of teeth and can prevent osteoporosis [91]. On the other hand, it has been reported that xylitol also exhibits anticancer properties by inducing apoptosis in these cells through the activation of the glutathione-degrading enzyme CHAC1, which in turn induces stress in the endoplasmic reticulum [92].

In another study, Baptista et al. reported the revalorization of previously hydrothermally treated vine shoots, enriched with grape pomace and wine lees for the production of xylitol and bioethanol through solid fermentation using a strain of *Saccharomyces cerevisiae*. In their research, they achieved titers of up to 37 g/L of xylitol and 50 g/L of bioethanol [93]. Bioethanol is a biofuel produced through the microbial fermentation of carbohydrates obtained from crops (first-generation bioethanol), lignocellulosic residues (second generation), and algae (third generation) [94]. It is an alternative to the use of fossil fuels, offering versatile production that projects future demand [95]. For their part, Garita et al. reported the use of vine shoots through liquid fermentation processes for the production of biobutanol using a strain of Clostridium beijerinckii reaching concentrations of up to 8 g/L [96].

### 4.3. Enzymes

Another interesting avenue for revalorization is the use of wine industry waste for the production of enzymes with various industrial applications. Authors [97] compared enzyme production using wheat bran and a combination of wheat bran with grape pomace (1:1 ratio), employing solid fermentation processes using a strain of *A. niger* 3T5B8. They reported the production of Xylanase (approximately 60 U/g 48 h), CMC (approximately 23 U/g 96 h) polygalacturonase (approximately 110 U/g 96 h) β glucosidase (approximately 100 U/g 96 h) and tannase (approximately 0.25 U/g 96 h) using conditions of 10 g of substrate, 60% humidity, inoculum of 1 × 10^7^ spores/mL and 37 °C. They found that the use of pomace contributed to the induction of the production of polygalacturonases and tannases, attributing this effect to the pectin content (4% in skin and 25% in seeds) as well as tannic acid present in the residue, also highlighting the fact that tannase production without the need for supplementation of the system with tannic acid [97]. Papadaki et al. evaluated the enzymatic production of *Pleurotus* sp. strains using grape pomace as a substrate through solid, semi-liquid, and submerged fermentation. They reported that laccase production was induced to a greater extent when working with a *P. ostruatus* strain through solid fermentation, reaching titers of 26,247 U/g under conditions of 4 g of substrate, 10 mL of liquid preinoculum at 26 °C, and 15 days of incubation. On the other hand, endoglucanase activity was produced to a greater extent using the submerged fermentation system, with titers of 0.93 U/g, using the same strain under conditions of 0.04 g/mL substrate, 10 mL of liquid preinoculum, 26 °C, 140 rpm, and 20 days of incubation [98]. These enzymes have interesting applications, with laccase being an enzyme applied in bioremediation processes, dyeing in the textile industry, paper pulp bleaching, biorefinery, wastewater detoxification, and the food industry, among others [99], while endoglucanase has applications in agriculture, paper industry, laundry animal feed, and food industry thanks to its effect of breaking β 1–4 glucosidic bonds. Revalorization of green tea waste through the production of cellulases. Filipe et al. carried out research to revalorize waste from the wine industry in combination with olive waste through solid fermentation processes with *A. niger* and *A. ibericus* strains. They evaluated different proportions of mixtures of raw olive pomace (COP), exhausted olive pomace (EOP), vine shoots (VS), and exhausted grape pome (EGP). They observed synergistic effects when combining the components and determined that the optimal mixtures were 30% EGP, 36% VT, and 34% EOP when working with A. ibericus, and 23% COP, 30% EGP, 33% VT, and 14% EOP with *A. niger*. Finally, by scaling up the process to the bioreactor level using the optimal substrate mixtures, they achieved enzymatic titers of 189.1 ± 26.7, 56.3 ± 2.1, and 10.9 ± 0.8 U/g for xylanase, cellulase, and β-glucosidase activities, respectively [100].

On the other hand, the revalorization of vine shoots through fermentation processes for enzyme production has also been reported. Hamrouni et al. worked on the revalorization of vine shoots in conjunction with other substrates (Wheat Bran, Olive Pomace, Oatmeal, Potato Flakes, and Olive Oil), evaluating the enzymatic production by solid fermentation processes using a strain of *T. asperellum*. They found that changing the proportion of the residue mixtures induced changes in the enzymatic titers obtained, the highest being 47.46 U/g dm for amylase activity, 19.01 U/g dm for lipase, 16.99 U/g dm for endoglucanase, and 9.97 U/g dm for exoglucanase [101].

Guimarães et al. conducted research evaluating rice husk, brewer’s spent grain, and vine shoot residues as substrates for the production of cellulases, xylanases, and amylases in solid fermentation using a strain of A. niger CECT 2088. They found that using only vine shoots as substrate produced enzymatic titers of 206 U/g for xylanases, 167 U/g for endoglucanases, 65 U/g for β glucosidases, and 48 U/g for amylases under fermentation conditions of 5 g of substrate, 75% humidity, 2 mL inoculum of a solution at a concentration of 1 × 10^6^ spores/mL, 25 °C, and 7 days of incubation in the dark [40].

Salgado et al. reported the production of cellulases, xylanases, and amylases in solid fermentation. Feruloyl esterases have been used to revalue residues from the wine and olive oil industries, reaching titres of up to 89.53 U/g using a strain of *A. niger* by solid fermentation processes. Feruloyl esterases are a subclass of carboxylic acid esterases, which can release phenolic acids such as p-coumaric and ferulic acids as well as their dimers. They also act with xylanases by breaking the diferulic bridges between the xylan chains, contributing to the release of lignin by opening these structures. These enzymes are used in saccharification processes of lignocellulosic materials, in the removal of lignin from cellulose in the paper industry, and in the extraction of ferulic acid, among other applications [90]. The main bioactive compounds and the fermentative process to extract them are summarized in Table 1.

## 5. Winery Residues in Pharmaceutical Industry Potential Applications

The winery residues have significant potential for applications in the pharmaceutical industry due to their reported health benefits. Particularly, polyphenols present anti-inflammatory, anticancer, antidiabetic, cardioprotective, and antioxidant activities. In pre-clinical studies, it has been reported that grape pomace reduces pro-inflammatory cytokines (IL-6, TNF α), improves gut microbiota balance, and inhibits NF-κB signaling [102].

In animal models, grape pomace was capable of reducing insulin resistance, obesity, and hepatic steatosis, thereby enhancing glucose tolerance and lipid metabolism [103,104]. Brings cardiovascular benefits, such as attenuation of hypertension and myocardial injury, attributable to nitric oxide modulation and antioxidant effects [103,104]. Diverse investigations demonstrate the high potential of winery by-products as ingredients for pharmaceutical or health applications, with valuable bioactive functions, such as mitigating oxidative stress, inflammation, and metabolic disorders [105].

## 6. Conclusions

In conclusion, the revalorization of wine industry waste through fermentation processes represents an innovative strategy that not only maximizes resource utilization but also significantly contributes to environmental and economic sustainability. Obtaining bioactive compounds from these wastes presents new opportunities for their application in various industries, including food, pharmaceuticals, and cosmetics. Chiefly, it opens the opportunity to obtain natural drugs for many diseases, from agricultural by-products or wastes. This approach aligns with the principles of the circular economy, promoting a model where waste is transformed into valuable inputs, thereby reducing environmental impact and encouraging more responsible resource use. Winery by-products are valuable functional ingredients with the potential to inhibit or prevent chronic diseases, including obesity, cancer, hypertension, diabetes, and atherosclerosis. Furthermore, this practice supports the Sustainable Development Goals of the UN 20-30 Agenda, particularly those related to responsible production and consumption, innovation, and climate action. The integration of these principles into the wine industry not only improves its sustainability but also strengthens its competitiveness in a market that is increasingly aware of the need for responsible and sustainable practices.

## Figures and Tables

**Figure 1 ijms-26-10820-f001:**
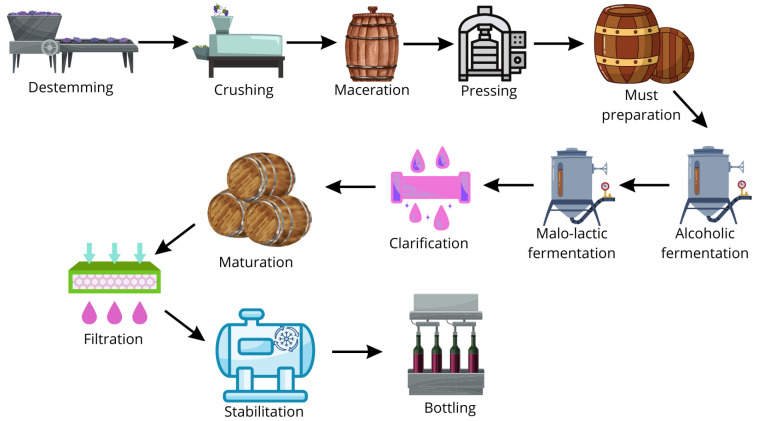
Main steps of winery industry.

**Figure 2 ijms-26-10820-f002:**
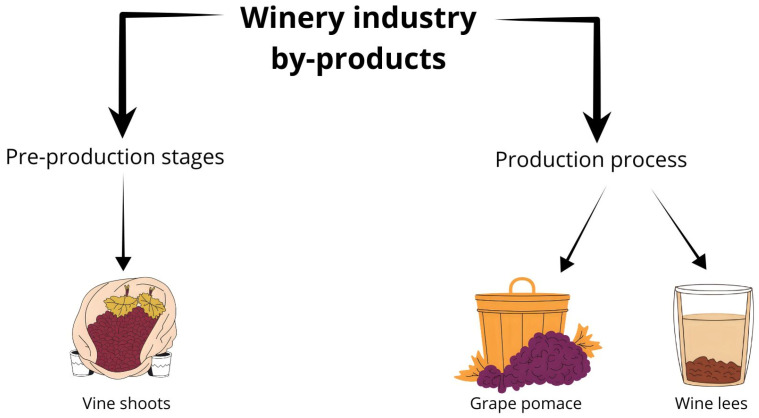
Winery industry by-products.

**Table 1 ijms-26-10820-t001:** Bioactive compounds, their uses, and the fermentative process for their production.

Bioactive Compound	Uses	Substrate	Fermentative Process
Polyphenolic compounds: -Anthocyanins	Alleviating metabolic dysfunction induced by obesity and diabetes, anticancer activity [75]	Grape pomace [56]	Solid and liquid fermentation (*Aspergillus niger* GH1) [56]
Alcohols: -Xylitol	Osteoporosis prevention, anticancer activity [91,92]	Vine shoots, grape pomace, and wine less [93]	Solid fermentation (*Saccharomyces cerevisiae*) [93]
Enzymes: -Laccase	Bioremediation processes, wastewater detoxification, and the food industry [99]	Grape pomace [98]	Solid fermentation (*Pleurotus ostruatus*) [98]

Table 1 shows only a one specific compound on different categories of bioactive natural drugs from winery wastes.

## Data Availability

No new data were created or analyzed in this study. Data sharing is not applicable to this article.

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
