# Peer review of "Wine Industry Waste as a Source of Bioactive Compounds for Drug Use"

_ijms, 2025, doi:10.3390/ijms262210820_

Round 1
Reviewer 1 Report
Comments and Suggestions for Authors
the manuscript needs a lot of reviews, the English must be reviewed. most of the sentences are so long. some corrections were cited on the text.

English language must be revised.
Author Response
Answers:
We really appreciate the comments and suggestions to improve the document.
- The first long sentence, indicated in the abstract section, page 1, lines 21-26 was rewritten and adjusted.
- Sentence in page 2, lines 42-46 was adjusted.
- Sentences on page 2, lines 47-59 were restructured.
- In page 3, line 80 Vitis viniferas was changed to Vitis vinifera (Italics).
- Sentences in page 3, lines 83-93 were adjusted.
- Long sentence in page 3, lines 104-110 was adjusted to have some short sentences.
- Page 3, line 120 Vitis vinifera was changed to Vitis vinifera.
- Page 4, line 124 V. viniferas was adjusted to vinifera.
- Sentence in page 4, lines 140-147 was adjusted.
- In page 6, line 218 was changed from Lactobacillus, Pediococcus, and Leuconostoc , to g Lactobacillus, Pediococcus, and Leuconostoc (not italic).
- Page 7, lines 275-279. As you kindly comment, the information placed in this section is expressed previously, so we delete this specific sentence.
- Page 7, lines 287, 292. The information is different and complementary to the previous information.
- Page 8, line 339. The word lees was adjusted to less.
- Page 9, line 343. The word lees was adjusted to less.
- Page 10, line 378. The word fungi was replaced by mould.
- References were adjusted to the journal style.
Reviewer 2 Report
Comments and Suggestions for Authors
The authors present a literature review on the utilization of wine industry by-products (grape pomace, seeds, lees, waste wine, process water) as a source of bioactive compounds with potential applications in pharmacology and other industrial sectors. As a key valorization process, the authors propose microbial fermentation as a central technology and highlight the importance of this approach in the context of the circular economy and the Sustainable Development Goals (SDGs).
Given that the management of agro-industrial waste is a rapidly developing field, and fermentation is aligned with both biotechnology and sustainability trends, the article can be considered innovative, timely, and above all of practical relevance.
Suggestions for improvement:
- The description of winemaking processes should be shortened, as it has a rather textbook-like character rather than a critical-scientific one, which would be more suitable for this type of article.
- Too little attention is devoted to the practical challenges of applying these by-products in the pharmaceutical industry, which is certainly an essential issue. A more critical discussion addressing potential limitations of the proposed solutions should therefore be added.
- The pharmacological aspect could be deepened by enriching the discussion with a review of clinical or toxicological studies.
- There is some repetitiveness of content—the introduction, global context, and conclusions partly duplicate information.
In my view, the novelty of the article lies in integrating existing knowledge into an interdisciplinary narrative that connects the wine industry (waste) with biotechnology and biopharmaceuticals, thereby aligning with the UN Sustainable Development Goals and the circular economy. For this reason, the paper represents a synthetic, interdisciplinary review rather than a breakthrough scientific report.
Author Response
Answers: We really appreciate your comments and suggestions that will improve the document.
- The description of winemaking processes was shortened to adjust the text like a critical-scientific paper.
- A critical discussion of preclinical and in vitro studies was added, particularly in the new section Winery residues in pharmaceutical industry potential applications. We would be delighted to include additional information from clinical studies; however, due to the current scarcity of available data, this has not been possible.
- We deepen in the information in the new section: Winery residues in pharmaceutical industry potential applications.
- The content of manuscript was revised to avoid repetitive information
Specific comments:
- Line 206, words Polpolyols,ntribuyting, were corrected to: Polyols and contributing
- The word wine less was adjusted in line 336 and also in the rest of the document for wine lees.
In table 1, Enzumes was changed to Enzymes
Reviewer 3 Report
Comments and Suggestions for Authors
The work is done with great care.
I found the introduction with to many historical details about the origin of wie production when the objective was wine industry waste

Author Response
Answer: We really appreciate your comments and suggestions that will improve the document.
- The description of winemaking processes was shortened to adjust the text like a critical-scientific paper decreasing the historical details about the origin of wine.
Round 2
Reviewer 1 Report
Comments and Suggestions for Authors
the authors have revised the article based on my comments.